# Nanoscale optical nonreciprocity with nonlinear metasurfaces

Aditya Tripathi[1], Chibuzor Fabian Ugwu[2], Viktar S. Asadchy[3,4], Ihar Faniayeu [5], Ivan Kravchenko[6], Shanhui Fan [3], Yuri Kivshar [1], Jason Valentine [2] & Sergey S. Kruk [1] ✉

Optical nonreciprocity is manifested as a difference in the transmission of light for the opposite directions of excitation. Nonreciprocal optics is traditionally realized with relatively bulky components such as optical isolators based on the Faraday rotation, hindering the miniaturization and integration of optical systems. Here we demonstrate free-space nonreciprocal transmission through a metasurface comprised of a two-dimensional array of nanoresonators made of silicon hybridized with vanadium dioxide ($VO_2$). This effect arises from the magneto-electric coupling between Mie modes supported by the resonator. Nonreciprocal response of the nanoresonators occurs without the need for external bias; instead, reciprocity is broken by the incident light triggering the $VO_2$ phase transition for only one direction of incidence. Nonreciprocal transmission is broadband covering over 100 nm in the telecommunication range in the vicinity of $\lambda = 1.5\,\mu$m. Each nanoresonator unit cell occupies only ~0.1 $\lambda^3$ in volume, with the metasurface thickness measuring about half-a-micron. Our self-biased nanoresonators exhibit nonreciprocity down to very low levels of intensity on the order of 150 W/cm² or a μW per nanoresonator. We estimate picosecond-scale transmission fall times and sub-microsecond scale transmission rise. Our demonstration brings low-power, broadband and bias-free optical nonreciprocity to the nanoscale.

Nanoresonators assembled into two-dimensional lattices—metasurfaces—enabled the miniaturization of functional optical components down to the nanoscale[1–3]. Over just a few years, we have observed impressive progress in both intriguing physics and important applications of resonant dielectric metasurfaces, ranging from fundamental concepts to mass-fabricated consumer products[4]. Passive and linear dielectric metasurfaces have started replacing conventional bulky optical components. A vital but relatively unaddressed problem of modern optics and subwavelength photonics is to achieve strong nonreciprocal optical response at the nanoscale.

In general, a nonreciprocal system exhibits different received-transmitted field ratios when their sources and detectors are exchanged[5,6]. The first experiments related to nonreciprocity in electromagnetism were performed by Faraday in 1845, and some of the first theoretical studies of associated phenomena were reported by Stokes in 1840, by Helmholtz in 1856, and by Kirchhoff in 1860[6]. Applications of nonreciprocity include realization of one-way propagation of light, such as in optical isolators and circulators. Most optical processes obey reciprocity, including refraction, diffraction, mode conversion, and polarization conversion. There exist three conceptual

[1]Nonlinear Physics Centre, Research School of Physics, Australian National University, Canberra, ACT, Australia. [2]Department of Mechanical Engineering, Vanderbilt University, Nashville, TN, USA. [3]Ginzton Laboratory, Department of Electrical Engineering, Stanford University, Stanford, CA, USA. [4]Department of Electronics and Nanoengineering, Aalto University, Espoo, Finland. [5]Department of Physics, University of Gothenburg, Gothenburg, Sweden. [6]Center for Nanophase Materials Sciences, Oak Ridge National Laboratory, Oak Ridge, TN, USA. ✉e-mail: sergey.kruk@outlook.com

pathways for breaking optical reciprocity: (i) materials exhibiting asymmetric permittivity/permeability tensors, (ii) time-varying systems, and (iii) nonlinear light−matter interactions. The dominant approach is based on materials with asymmetric tensors, such as ferrites[7,8]. However, ferrite-based systems are not compatible with nanotechnology as they rely on rather large permanent magnets or resistive/superconductive coils. The second approach based on time-varying systems[9–11] has enabled the miniaturization of nonreciprocal components down to the microscale[12,13], however, it imposes major technological challenges for a further miniaturization to the nanoscale due to its weak response, power inefficiency, and overall complexity of the modulation required to operate in the optical spectral domain.

This suggests that at present, the most feasible pathway towards nonreciprocity at the nanoscale is via nonlinear light-matter interactions. Nonlinearity-induced nonreciprocity comes with fundamental limitations[6,14], such as the inability to operate under two or more simultaneous excitations. Nonlinear nonreciprocity exists only within a certain range of incident powers, and it may have a trade-off between the range of operation powers and insertion loss[15]. On the other hand, several aspects of nonlinear nonreciprocity can be advantageous. Nonlinear nonreciprocity is self-induced, and therefore it can be implemented in fully passive optical systems. Nonlinear nonreciprocal components do not require any external biases, thus being much simpler and easier to miniaturize than their magneto-optical or time-variant counterparts. Several optical applications of nonlinearity-induced nonreciprocity may benefit from these advantages, including optical switches, asymmetric power limiters, and LiDARs.

Nonlinearity-induced nonreciprocity at the sub-micrometer scale has been studied in unstructured thin films[16–19]. However, such observations were accompanied by low levels of transmission and high insertion losses, which hindered their development beyond initial proof-of-concept experiments.

Nonlinear nonreciprocity at the micrometer scale has been studied in guided platforms of waveguides and ring resonators[20,21]. Parity-time symmetry in waveguiding systems with loss and gain was employed to enhance nonlinearity-based nonreciprocity[22–24]. However, such waveguiding platforms cannot be miniaturized to the sub-wavelength scale.

A promising pathway towards nonreciprocity at the nanoscale is brought about by nanoresonators with carefully engineered geometries. In contrast to unstructured thin films, nanoresonators are capable of boosting the efficiencies of nonlinear light–matter interactions by orders of magnitude[25,26]. Symmetry breaking in optical systems due to nonlinearity has recently been demonstrated in the parametric generation of optical harmonics resulting in the asymmetric formation of topological edge states[27] and asymmetric generation of optical images[28].

Recently, there has been an interest in theoretical studies of various nonlinear nanostructures for asymmetric and nonreciprocal light control[29–35]. Plasmonic metasurface with asymmetries in nonlinear (third harmonic) light generation were demonstrated experimentally[36]. Silicon grating-like metasurfaces hosting high-Q resonances were demonstrated experimentally to exhibit non-reciprocal transmission via the intrinsic intensity-dependent response of silicon[37], albeit only for high levels of incident power (mega-Watts per cm²) and over a narrow spectral range (nanometers).

Optical nonreciprocal responses demonstrated to date rely on components substantially larger than the wavelength of light in at least two spatial dimensions, including the systems reliant on collective effects in phase-gradient[36], and grating-like implementations[37].

## Results

Here, we experimentally demonstrate a half-a-micron-thick non-reciprocal metasurface with different forward and backward transmission (see Fig. 1). The enabling physics behind our demonstration is the realization of magnetic and electric Mie resonances via the engineering of nanoscale geometry of the resonators. Nonreciprocal effects arise from the magnetoelectric coupling between the resonant modes. The metasurface consists of nanoresonators made of silicon (Si) placed on a thin $VO_2$ film over a glass substrate and embedded into PMMA, as shown in Fig. 1b. PMMA and glass create a nearly homogeneous and isotropic optical environment for the Si−$VO_2$ nanostructure. A few nanometer thin encapsulation layer of $Al_2O_3$ is placed between the $VO_2$ and the Si disks for $VO_2$ protection. Si disks also have a few nanometers thin $Al_2O_3$ top caps which are the residues of our fabrication process that uses $Al_2O_3$ as a hard mask for Si etching (described below). $Al_2O_3$ contribution to the metasurface's optical properties is negligibly small due to its thinness. $VO_2$ is a phase-transition material whose crystalline structure can be changed by changing its temperature[38]. At room temperature, the $VO_2$ features monoclinic crystalline lattice, and it acts as an insulator at optical frequencies. At around 68 °C, it transitions to a tetragonal crystalline lattice, and it acts as a conductor. $VO_2$ is a particularly attractive phase-transition material whose dynamical change of phase corresponds to a subtle crystalline-to-crystalline transition and is therefore fully reversible. The exceptionally large complex refractive index variation produced by the insulator-to-conductor transition of this material made it an attractive choice for metasurfaces reconfigurable thermally or electrically[39–44]. $VO_2$ insulator-to-conductor phase transition was demonstrated to occur on a picosecond scale, paving the way to ultra-fast applications[45]. We note that while here we focus on the $VO_2$ material, the principles of operation of our metasurface should be immediately applicable to other types of phase-change/phase-transition materials[38] notably including GST materials in which femtoseconds scale switching times have been reported[46].

### Computational design

We design the metasurface to have high transmission in the 1.4−1.6 μm wavelength range when the $VO_2$ is in its insulating phase (Fig. 2a, black line). The metasurface consists of silicon disks 540 nm in height and diameter residing on 35-thin $VO_2$ film arranged into a square lattice with 820 nm period (see sketch in Fig. 1b). The asymmetric design of the metasurface along the optical axis (due to the presence of the $VO_2$ film) leads to asymmetric absorption of light by the metasurface (Fig. 2a) for forward and backward incidence. To this end, the described functionality is reciprocal, and the transmission of the metasurface remains the same for the opposite directions of propagation as the differences in absorption are compensated by differences in reflection. However, difference in absorption for forward and backward directions leads to differences in the temperature of the $VO_2$ film (Fig. 2b). Heating in its turn may lead to the phase transition of $VO_2$ to its conductive phase, which in our design drastically reduces the metasurface transmission (see Fig. 2c, d). In our calculations, we exemplarily excite the metasurface with a 100 ps pulse such that the $VO_2$ heating for the backward direction is fully sufficient for the phase transition to occur, while heating for forward direction remains insufficient under the same excitation conditions. In this setting, transmission fall time is comparable to the incident excitation pulse of 100 ps, which agrees with experimental observations of insulator-to-conductor transition times of $VO_2$ films with fall times on the order of 26 ps[45]. At the same time, transmission modulation in the forward direction remains minor. The $VO_2$ film then cools down to room temperature over a time period of about 1 μs. However, the transmission rise occurs on a faster scale, on the order of 100 fs (from the end of the excitation pulse to 90% of total rise). This demonstrates that the metasurface is nonreciprocal under picosecond pulse excitation with up to 1 kHz pulse repetition rate. In addition, we performed similar calculations for continuous-wave (CW) excitation (see Supplementary) and estimated the transmission rise and fall times to be on the order of 5 and 23 microseconds. We attribute the difference in the dynamics to

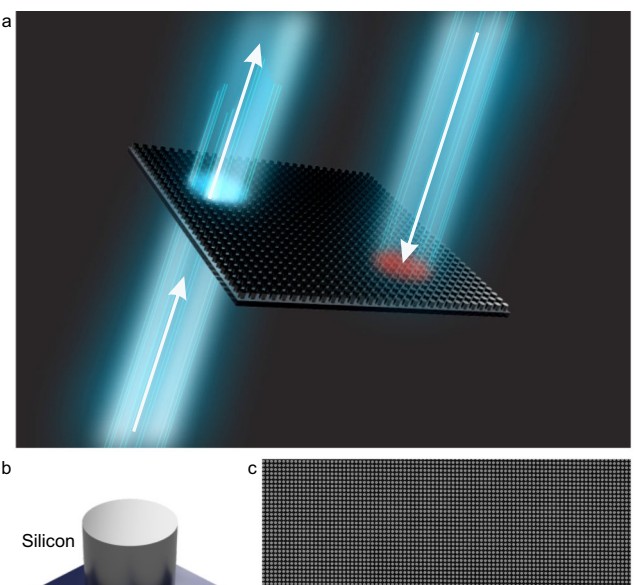

**Fig. 1 | Nonreciprocal transmission of light through a hybrid Si–VO₂ metasurface. a** Concept image of one-way transmission through a metasurface. **b** Schematics of a subwavelength resonator (metasurface unit cell): silicon disk placed on top of VO₂ film. The metasurface resides on glass and is covered with polymethyl methacrylate (PMMA) with a refractive index similar to glass. This creates an environment close to homogeneous and isotropic for the Si–VO₂ nanostructures. **c** Electron microscope images of the fabricated Si–VO₂ metasurface.

the following. For the case of the pulsed excitation (e.g., 100 ps pulses, 1 kHz repetition rate), the pulse heats up only the VO₂ film causing only small changes in the temperature of the surrounding materials (SiO₂, Al₂O₃, PMMA). In the case of CW excitation, the temperature distribution reaches a steady state in the materials in the immediate vicinity of the VO₂ film, resulting in a much larger thermal mass being heated.

Numerical simulations of the spectral and temporal response of the metasurface were performed in Comsol Multiphysics® using the Floquet periodic boundary conditions. Geometrical parameters of the metasurface including the VO₂ thickness were optimized to maximize (i) transmission in the forward direction and (ii) isolation in the backward direction. We utilized the Finite Element Method (FEM) with Comsol Multiphysics® software to analyze the thermal and optical characteristics of the proposed VO₂ metasurface. Port boundary conditions facilitated the launch of transverse magnetic (TM) or transverse electric (TE) polarized plane waves, while periodic boundary conditions simulated the side boundaries. Although lasers typically exhibit Gaussian beam profiles, the simulation geometry's small size enabled us to treat the laser as a plane wave. By combining the Heat Transfer in Solids and Electromagnetic Waves modules, we modeled transient heating and calculated transmittance over time. Specifically, we examined the transmission under front and back side illumination at 1.44 μm wavelength, with an input power intensity of 22.3 kW/cm² and a pulse duration of 100 ps (assuming a simplistic rectangular pulse shape). The initial temperature was set to room temperature (293 K) and was uniform throughout the structure. Electromagnetic pulse in COMSOL introduced a heat source term based on optical absorption of materials calculated within the Electromagnetic Waves module.

Temperature-dependent thermal conductivity, heat capacity, and density values of materials PMMA, Al₂O₃, and SiO₂ in the thermal simulations were taken from experimental data[47–49]. Meanwhile, the VO₂ dielectric constant was determined using the Bruggeman effective medium theory[50]

$$\epsilon_{VO_2} = \frac{1}{4}\left[\epsilon_i(2-3V) + \epsilon_c(3V-1) + \sqrt{\left[\epsilon_i(2-3V) + \epsilon_c(3V-1)\right]^2 + 8\epsilon_i\epsilon_c}\right] \quad (1)$$

the $\epsilon_i$ and $\epsilon_c$ are represented by the dielectric constants of the insulating and conducting phases of VO₂, respectively. They have been measured experimentally. The metallic volume fraction, indicated by V, can be calculated using the following formula:

$$V = 1 - \frac{1}{1 + e^{\frac{T-T_c}{\Delta T}}} \quad (2)$$

$T$ is the ambient temperature, $T_c$ is the critical temperature of VO₂, and $\Delta T$ denotes the transition width and equals 2K[51]. The optical properties of VO₂ are incorporated into an electromagnetic simulation to analyze the metasurface's time-dependent transmittance and temperature. The transmission coefficient was calculated from S-parameters and temperature was integrated over the volume of the VO₂ film.

Metasurfaces governed by lower-order Mie resonances typically exhibit tolerance to oblique incidence angles of up to a few degrees[52]. We estimate that our metasurface demonstrates small changes of transmission for ±5-degree incident angle variation with substantial changes for ±10-degree variation (Supplementary Fig. S8).

## Theoretical description

The functionality of the metasurface arises from the resonant scattering of its individual nanoresonators. We perform decomposition of the total scattering into a series of Mie multipoles. The multipolar decompositions were evaluated as described in ref. 53. In the absence of the VO₂ film, the metasurface response is dominated by only the ED and MD which are balanced at around 1465 nm wavelength (see corresponding spectra in Supplementary Fig. S2 and the multipolar balance at 1465 nm wavelength in Fig. 2e). In this case the multipolar composition, and therefore the optical response, is the same for forward and backward directions.

The presence of the VO₂ film breaks the geometrical symmetry introducing contributions to scattering from higher-order multipoles: electric quadrupole (EQ) and magnetic quadrupole (MQ) (see multipolar spectra in Supplementary Fig. S1 and multipoles' amplitudes at around 1470 nm wavelength in Fig. 2e).

The VO₂-induced asymmetry of the design enables magnetoelectric coupling between the Mie multipoles resulting in different compositions for the forward and backward- directions. High transmission of the metasurface is enabled by the balance of symmetric (ED, MQ) and anti-symmetric multipoles (MD, EQ) which interfere constructively in the forward direction and destructively in the backward direction, resembling the conditions of generalized Huygens' principle[54]. Generalized Huygens metasurfaces are known for their extended spectral ranges of operation[54]. Here we estimate the operation range of about 100 nm which we define as >10 dB isolation as per Fig. 2e. Prevalence of magnetic multipoles for the backward direction of excitation is associated with higher field concentration in the VO₂ material and higher absorption (see Supplementary Fig. S6). Metasurfaces that exhibit magnetoelectric coupling show peculiar photonic functionalities such as polarization transformations[55,56], including asymmetric transmission[57,58], asymmetric reflection[59,60], transverse Kerker effect[61,62], photonic analogs of spin-Hall effects[63], photonic Jackiw–Rebbi states[64] and nontrivial topological phases[65].

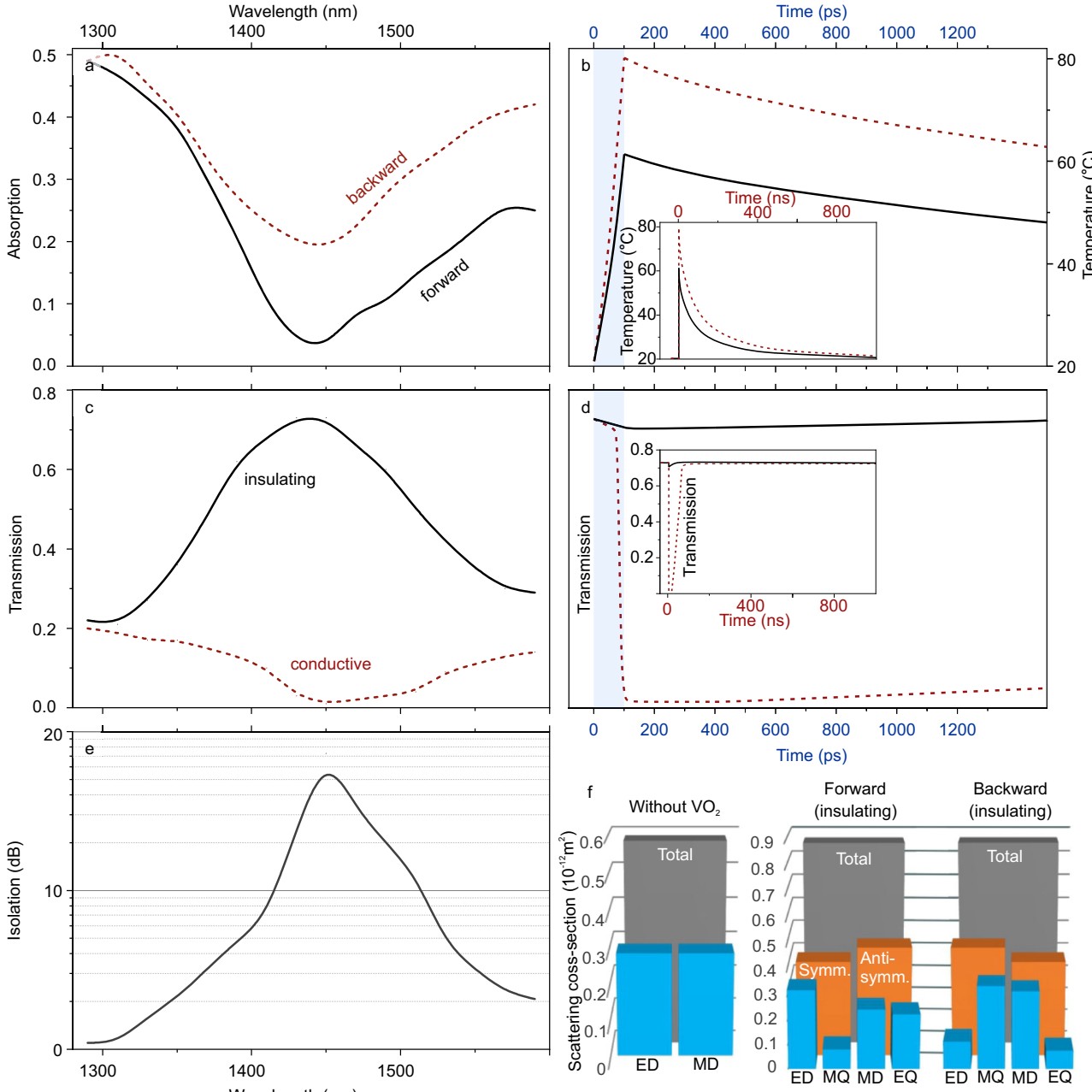

**Fig. 2 | Theoretical study of the spectral and temporal response of the non-reciprocal metasurface. a** Absorption spectra of a Si–VO₂ metasurface for the insulating VO₂ phase and two directions of incidence (forward/backward). **b** Temporal dynamics of VO₂ temperature for the two opposite directions of excitation at 1450 nm wavelength under an excitation with 100 ps pulse. **c** Transmission spectra of the metasurface for insulating (black, solid) and conductive (red, dashed) VO₂ phases. **d** Temporal dynamics of the metasurface

transmission for the two opposite directions of excitation at 1450 nm wavelength and under an excitation with 100 ps pulse. **e** Contrast between transmission in insulating and conductive phases. **f** Multipolar composition of the unit cell scattering: (left) metasurface without the VO₂ film featuring identical scattering for the forward and backward directions, (right) with the VO₂ film for the two opposite directions of excitation. Here ED electric dipole, MD magnetic dipole, EQ electric quadrupole, MQ magnetic quadrupole.

To provide a qualitative picture explaining the reason for the asymmetry in multipolar content for different illumination directions shown in Fig. 2c, we turn to the classical multipolar theory. The four lowest multipolar moments (electric **p** and magnetic **m** dipoles as well as electric $Q^e$ and magnetic $Q^m$ quadrupoles) induced in the resonator by general electric $\mathbf{E}(\omega,\mathbf{r})$ and magnetic $\mathbf{H}(\omega,\mathbf{r})$ field distributions are given by[66,67]

$$p_i = \alpha_{ij}^{ee} E_j + \alpha_{ij}^{em} H_j + \gamma_{ijlm} E_j k_l k_m + \ldots \qquad (3)$$

$$m_i = \alpha_{ij}^{mm} H_j + \alpha_{ij}^{me} E_j + \eta_{ijlm} H_j k_l k_m + \ldots \qquad (4)$$

$$Q_{ij}^e = \beta_{ijk}^{ee} E_k + \beta_{ijk}^{em} H_k + \ldots \qquad (5)$$

$$Q_{ij}^m = \beta_{ijk}^{mm} H_k + \beta_{ijk}^{me} E_k + \ldots \qquad (6)$$

Here, we used the Einstein index notation, $\alpha_{ij}^{ee}$, $\alpha_{ij}^{mm}$, $\alpha_{ij}^{em}$, and $\alpha_{ij}^{me}$ are the second-rank polarizability tensors, and the latter two are

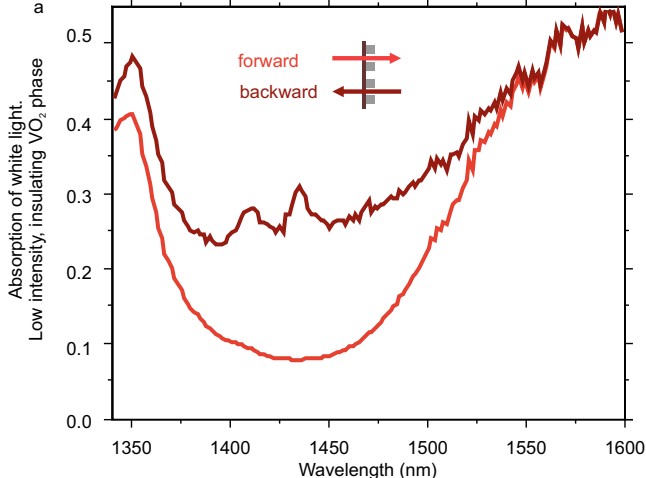

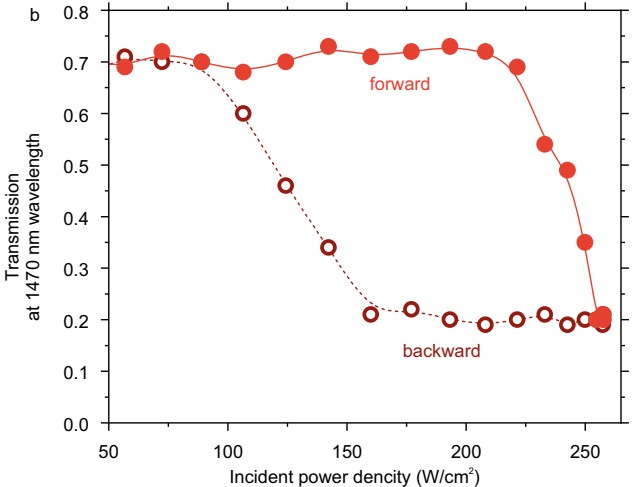

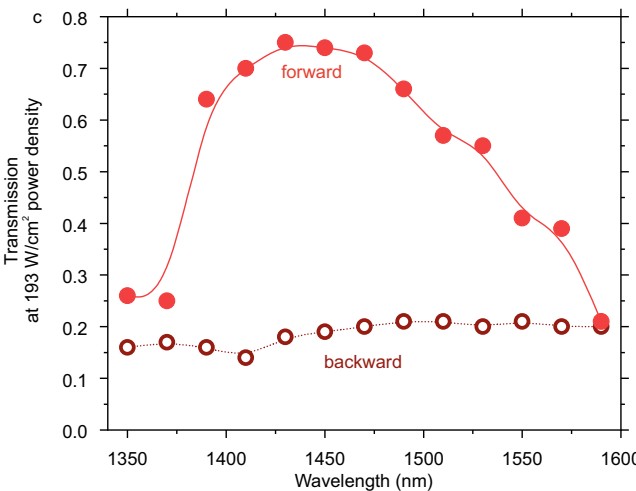

**Fig. 3 | Experimental demonstration of nonreciprocal transmission of light with a hybrid Si–VO₂ metasurface. a** Experimental white-light absorption for two opposite directions of illumination. **b** Experimental transmission for two opposite directions of illumination at 1470 nm wavelength as a function of an increasing intensity of light. For power densities in the range 150–250 W/cm², the metasurface demonstrates pronounced contrast between forward and backward transmissions. **c** Transmission for two opposite directions of illumination vs. wavelength at 193 W/cm² power density.

related to bianisotropy (dipolar magnetoelectric coupling)[68]. The third-rank tensors $\beta^{\mathrm{ee}}_{ijk}$, $\beta^{\mathrm{mm}}_{ijk}$, $\beta^{\mathrm{em}}_{ijk}$, and $\beta^{\mathrm{me}}_{ijk}$ are referred to as quadrupole polarizabilities[66] with the latter two being magnetoelectric quadrupolar polarizabilities. Finally, the third-rank tensors $\gamma_{ijlm}$ and $\eta_{ijlm}$ are referred to as hyperpolarizabilities[69] and merely represent polarization effects induced by the higher-order elements in the Taylor series of the incident fields $E_i$ and $H_i$ with respect to wave vector $k_i$. In our qualitative analysis, we assume that the resonator size is sufficiently small compared to the wavelength so that only the first two terms in Eqs. (1)–(4) can be considered as significant. Next, without loss of generality, we consider that incident light on the resonators propagates along the $\pm z$-direction with electric and magnetic fields having $E_x$ and $\pm H_y$ as the only nonzero components. We further took into account the $C_{4v}$ point group symmetry of the unit cell and assumed that the metasurface obeys optical reciprocity in the linear regime. This further reduces the number of polarizability terms: $\alpha^{\mathrm{me}}_{yx} = -\alpha^{\mathrm{em}}_{xy}$. The Eqs. (1)–(4) then take the following forms:

$$p_x = \alpha^{\mathrm{ee}}_{xx} E_x \pm \alpha^{\mathrm{em}}_{xy} H_y, \qquad (7)$$

$$m_y = \pm \alpha^{\mathrm{mm}}_{yy} H_y - \alpha^{\mathrm{em}}_{xy} E_x \qquad (8)$$

$$Q^{\mathrm{e}}_{xz} = \beta^{\mathrm{ee}}_{xzx} E_x \pm \beta^{\mathrm{em}}_{xzy} H_y, \qquad (9)$$

$$Q^{\mathrm{m}}_{yz} = \pm \beta^{\mathrm{mm}}_{yzy} H_y + \beta^{\mathrm{me}}_{yzx} E_x \qquad (10)$$

The double sign in the equations denotes the scenario of two opposite light illuminations. All the other projections of the dipole and quadrupole moments (e.g., $p_y$ and $m_x$) as are assumed to be negligible due to the cylindrical symmetry of the resonator. From Eqs. (5) and (6), one can deduce that the contrast in the dipole moment strengths $|p_x|$ and $|m_y|$ for the forward and backward illuminations stems from their bianisotropic response ($\alpha^{\mathrm{em}}_{xy} \neq 0$, $\alpha^{\mathrm{me}}_{yx} \neq 0$). Indeed, the asymmetric (due to the VO₂ film) resonator shown in Fig. 1b is known to exhibit bianisotropy[41], which explains why it exhibits different induced dipole moments for opposite illuminations as seen from Fig. 3d. Without the VO₂ film, the substrate-induced bianisotropy does not occur, which confirms the absence of the contrast for the same moments in Fig. 2c. Furthermore, the contrast in the quadrupole moment amplitudes $|Q^{\mathrm{e}}_{xz}|$ and $|Q^{\mathrm{m}}_{yz}|$, seen in Fig. 3d, according to Eqs. (5) and (6) stems from the nonzero quadrupolar magnetoelectric coupling ($\beta^{\mathrm{em}}_{xzy} \neq 0$, $\beta^{\mathrm{me}}_{yzx} \neq 0$).

Thus, the dipolar and quadrupolar magnetoelectric coupling leads to different multipolar compositions for the opposite directions of excitation, particularly for the high contribution of ED for forward excitation and for the high contribution of MQ for backward excitation. Although the different multipolar compositions lead to the same transmissions for the two opposite illuminations, they result in different absorptions. MQ mode is more tightly localized spatially leading to higher field concentration inside the VO₂ film, and thus to higher absorption. Absorption of light in the VO₂ film increases its temperature, and enough light-induced heating triggers a phase transition from the insulating to the conductive phase.

## Nanofabrication

To fabricate the metasurface, 35 nm of VO₂ films were grown on a fused silica wafer and annealed in 250 mTorr of oxygen at 450 °C. 10 nm of aluminum oxide (Al₂O₃) serving as a spacer and etch stop layer was deposited on the VO₂ via e-beam evaporation. In all, 540 nm thick amorphous silicon was grown on the Al₂O₃–VO₂ layered structure. The resonator structure was created by a standard electron beam lithography process with a PMMA photoresists and a 1:3 MIBK/IPA developer. An Al₂O₃ hard etch mask was prepared by electron beam evaporation, and the undeveloped resist was successfully lifted off in

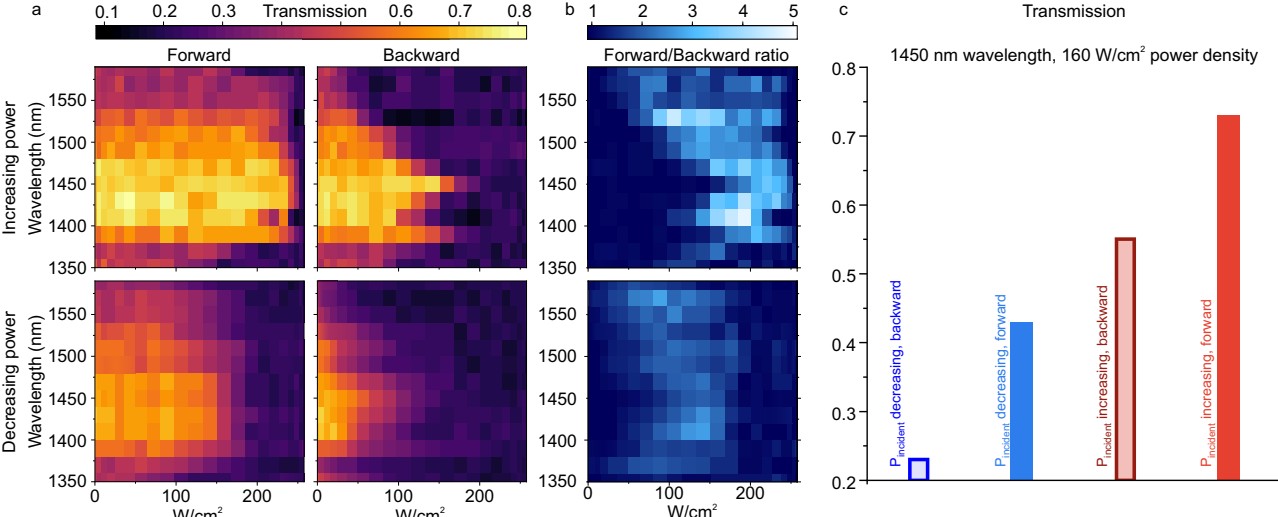

**Fig. 4 | Experimental demonstration of the interplay between nonreciprocity and optical bistability. a** Experimentally measured maps of transmission as functions of the intensity and wavelength of incident light. Top row: light intensity is increasing. Bottom row: light intensity is decreasing. First column: for the forward illumination; second column: for the backward illumination. Differences in transmission between the rows are caused by the bistability and differences between the first two columns are caused by nonreciprocity. **b** Forward-to-backward ratio showing nonreciprocity over a range of wavelengths and intensities for both cases of the increasing and decreasing light intensity. **c** Four distinct transmission levels for the same incident wavelength and intensity enabled by the interplay of non-reciprocity and bistability.

an acetone bath. The samples underwent reactive ion etching (RIE) to create silicon nanopillars. Finally, a top layer of PMMA was spun onto the sample to create an index-matched layer. An electron microscope image of the fabricated metasurface is shown in Fig. 1c.

## Optical experiments

We proceed with optical diagnostics of the fabricated metasurfaces. In Fig. 3a, we show white-light measurements of the metasurface absorption for the insulating phase of the $VO_2$ for forward and backward scenarios of illumination. The absorption is derived from transmission and reflection measurements. Transmission through the sample is referenced to spectra obtained through PMMA-coated glass substrate (we note that PMMA refractive index is similar to that of glass) and further normalized to its estimated value with reference to air. Reflection from the sample is referenced to reflection of an uncoated gold mirror and further renormalized to 100% reflective mirror. This observation agrees with Fig. 2b. Next, we illuminate the metasurface with a tunable continuous-wave (CW) diode laser with power less than 10 mW. A small portion of the laser beam is reflected onto an Ophir power meter which monitors the intensity level. The power is attenuated with a set of polarizers. The laser beam is weakly focused (spot size ~200 microns) onto the metasurface with a long-focal distance lens ($f = 200$ mm achromatic doublet). Given the output laser beam radius of about 1 mm, the numerical aperture of the focusing beam is NA = 0.005, thus the excitation condition is close to a plane-wave illumination. We use a field diaphragm to detect signal from the metasurface sample only. Then we detect light transmitted through the metasurface with a second Ophir power meter. For forward/backward experiments, we flip the sample inside the setup. Figure 3b, c shows transmission through the metasurface for forward and backward scenarios of excitation normalized here to the transmission through the PMMA-coated glass substrate. We perform a set of test experiments at room temperature as well as at a biased 40 °C and 60 °C temperatures monitored by a controller Thorlabs TC300 (see details in Supplementary Fig. S3). We observe nonreciprocal behavior of the metasurface for all the temperature biases at similar levels of incident power. We further choose to work at 40°C temperature which requires 30% less incident power to trigger the $VO_2$ transition

compared to room temperature while keeping the sample sufficiently far from material hysteresis of the $VO_2$ film.

We experimentally observe pronounced nonreciprocal effects in transmission that resemble closely theoretical calculations. We attribute discrepancies between theory and experiment to imperfections associated with nanofabrication of the VO2 and silicon-based nanostructures. The use of reactive ion etching likely causes some damage in the $VO_2$, reducing contrast in switching. This could be avoided by employing an architecture where $VO_2$ is deposited after patterning of the silicon resonator layer.

We finally study the performance of the metasurface under the increasing and decreasing intensities of the incident light beam (thus for heating and cooling of the $VO_2$). Vanadium dioxide is known to exhibit hysteresis behavior for heating/cooling cycles. In our optical experiments, material hysteresis leads to optical bistability, where the transmission becomes dependent on the previously applied level of intensity (higher/lower). Bistability, combined with nonreciprocity, thus results in four different transmission dependencies of the metasurface: for forward/backward and for increasing/decreasing light intensity (see Fig. 4a). For both the increasing and the decreasing levels of intensity the metasurface shows pronounced nonreciprocal behavior over a range of intensities and wavelengths.

Figure 4c shows a peculiar example of four distinct levels of transmission at the same wavelength and the same incident intensity depending on the combination of two factors: direction of the excitation (forward/backward) and previous level of intensity (higher/lower).

## Discussion

In summary, we have demonstrated a high contrast between forward and backward transmission of light through Si–$VO_2$ hybrid metasurfaces of a subwavelength thickness. Nonreciprocal transmission is enabled by a phase transition of the $VO_2$ material acting as a strongly nonlinear self-biased medium. The basic principles of operation of our nonreciprocal metasurface should be immediately applicable to other types of phase-change/phase-transition materials. Our metasurface can operate at low levels of light intensities of the order of 100 W/cm² of continuous-wave excitation. This is in striking contrast with typical

nonlinear Kerr-type self-action devices in nanophotonics[36,37] often requiring the pulsed peak powers reaching and exceeding the values of GW/cm². We estimate fast switching times of the metasurface enabled by the picosecond-scale insulator-to-metal transition of the $VO_2$. We believe this type of nonlinear metasurface can pave a way towards nonreciprocal nanoscale components capable of functioning at low levels of incident power. Our hybrid metasurface demonstrates over 100 nm bandwidth in the vicinity of 1.5 μm wavelength. Optical non-reciprocity originates from the response of a single resonator/unit cell of a subwavelength volume. This opens up an untapped potential for the design freedom of functional nonreciprocal metasurfaces assembled from dissimilar resonators for asymmetric control of light[28]. Nonreciprocal passive flat optics could dramatically advance many applications including machine vision, photonic information routing, and switching.

## Data availability

The data generated in this study have been deposited in the Figshare database under the accession code https://doi.org/10.6084/m9.figshare.25609842.v1. Additional information will be provided by S.K. on request.

## Code availability

The modeling was performed with commercial software Comsol Multiphysics®. Additional information will be provided by A.T. and I.F. on request.

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

## Acknowledgements

S.K. acknowledges financial support from the Australian Research Council (grant DE210100679) and from the EU Horizon 2020 Research and Innovation Program (grant 896735). Y.K. acknowledges financial support from the Australian Research Council (grant DP210101292), International Technology Center Indo-Pacific (ITC IPAC), and Army Research Office (contract No. FA520923C0023). V.S.A. acknowledges the Academy of Finland (Project No. 356797), the Finnish Foundation for Technology Promotion, and Research Council of Finland Flagship Programme, Photonics Research and Innovation (PREIN), decision number 346529, Aalto University. J.V. acknowledges support in a portion of nanofabrication that was conducted as part of a user project at the Center for Nanophase Materials Sciences (CNMS), at the US Department of Energy, Office of Science User Facility at Oak Ridge National Laboratory. S.F. acknowledges the support of a MURI project from the U.S. Air Force Office of Scientific Research (AFOSR) (Grant No. FA9550-21-1-0312).

## Author contributions

A.T. and S.K. conceived the idea, A.T. performed electromagnetic computer simulations, I.F. performed joint electromagnetic and thermal computer simulations, V.S.A., S.F., and Y.K. contributed analytical theory, C.F.U., I.K., and J.V. fabricated the sample, C.F.U. and J.V. performed initial characterization of the sample, A.T. and S.K. performed optical experiments, all co-authors contributed to discussions of the results and manuscript writing.

## Competing interests

The authors declare no competing interests.
