## [Peer Review File · Nature Communications]

REVIEWER COMMENTS

Reviewer #1 (Remarks to the Author):

The authors demonstrate nonreciprocal transmission through an ultrathin (half-a-micron-thick) silicon metasurface hybridized with an even thinner (35nm thick) vanadium dioxide (VO₂) layer. Reciprocity is broken by VO₂ undergoing a phase transition from its insulating to conductive phase. Hence, the nonreciprocal effect is based on the thermal phase transition of VO₂ boosted by the Mie resonant response of the dielectric metasurface. The phase transition is induced all-optically by the incident light, and it is designed to occur under different conditions for the opposite directions of illumination.

The authors have adequately responded to the majority of the previous reviewers' comments. While it is true that some parts of theory and relevant nonreciprocal concepts have been presented in other works, I strongly agree with the reviewer's 2 opinion that the paper should be published after some revisions. The experimental demonstration of such ultrathin nonreciprocal nanoscale structure with exceptional nonreciprocal transmission performance is unique and would be very interesting to the broad audience of Nature Communications.

1) Such broadband, low power, and nanoscale ultrathin nonreciprocal design is commendable and unique feature of this work. In addition, the effect happens in a self-biased way, i.e., by the signal itself travelling through its structure. Hence, this modulation technique is more simple to implement compared to magnets or time-modulation and requires reduced power. I suggest all these major advantages of the presented system to be further stressed in the revised paper, even in the abstract section, which currently seems to be too technical for a broad audience paper.

2) Since the presented thermal nonreciprocity occurs in a relatively slow time (on the order of 1 millisecond), I suggest the authors to consider adding a simulation plot similar to figure 5b in ref [34] (10.48550/arxiv.2210.05586) that demonstrates how the temperature is increased in the sample as a function of time. I was also wondering if there is a way to either experimentally or theoretically demonstrate how the nonreciprocal transmission varies as a function of time in the case of front and back side illumination during the phase transition of VO₂.

3) Relevant to my previous comment, a theoretical or experimental plot that demonstrates the nonreciprocal transmission variation in the case of front and back side illumination but, now, as a function of temperature will be very interesting. It will demonstrate what happens during the phase transition of VO₂, similar to the theoretical plot (fig. 3a) in this paper: Optics Express Vol. 30, Issue 22, pp. 39716-39724 (2022)

4) The paper presents a passive nonlinear-based self-induced nonreciprocal system. However, efforts are also underway to further improve the performance of these systems by incorporating active (gain) media combined with nonlinear effects. As an example, active asymmetric systems based on the coexistence of nonlinearity and parity–time symmetry can further improve the nonreciprocal transmission contrast (check for example the following papers: i) Nature Phys. 10, 394–398 (2014), ii) Nature Photonics volume 8, pages524–529 (2014), iii) Adv. Opt. Mater. 7, 1901083 (2019).

I suggest to add some comments in the introduction relevant to active nonreciprocal nonlinear systems.

5) Please doublecheck the entire paper and supplementary for typos. As an example, the units (W/cm²) are missing in the x-axis of Figure S4 in the supplementary document.

Reviewer #2 (Remarks to the Author):

The manuscript entitled “Nanoscale optical nonreciprocity with nonlinear metasurfaces” by Aditya Tripathi et. al. presented an a-Si on VO₂ thin film nonreciprocal metasurface leveraging the out-of-plane anisotropic resonance scattering to induce different modal overlap with the absorptive VO₂ thin film between the forward and backward propagating beams. The absorption further induced the VO₂ phase-transition to metallic state (nonlinearity), achieving nonreciprocal light transmission.

My biggest concern is lack of novelty. The concept is well known and there are already several reports on such a-Si/VO₂ hybrid structure. Some examples are:

Photon. Res. 10, 373-380 (2022),

Conference on Lasers and Electro-Optics, Technical Digest Series (Optica Publishing Group, 2022), paper FTh4D.1.

An advancement to the previous work done by the authors’ group (ACS Photonics 2021, 8, 4, 1206–1213) is merely changing the phase-transition stimulus from direct heating to laser absorption. The structure, and even the results are quite similar.

Next, the authors failed to present a convincing implementation scenario of such structure. While they stated in the manuscript LiDARs, optical switches are possible. LiDARs are a bi-directionally transmitting case where such nonlinearity induced nonreciprocity fails. They also require phase manipulation, an important functionality this simple meta-structure does not have. Optical switching of ms speed is also way too slow for high-speed routing.

In addition, the performance of this device is very limited. For both on-chip and free space nonreciprocal transmission, more than 10 dB isolation ratio are reported. However, in this work, the largest ratio is around 5. Though theoretically predicted a ratio of 80, its huge difference with experimental demonstration are not discussed.

Overall, due to insufficient novelty and limited advancement compared to previous works, I cannot recommend its publication in Nature Communications.

Reviewer #3 (Remarks to the Author):

Title: Nanoscale optical nonreciprocity with nonlinear metasurfaces

Manuscript # NCOMMS-23-27507-T

A. Tripathi et.al., have explored a metasurface made of Silicon nanopillars supporting Mie resonances. Thin film VO₂ exhibiting reversible insulator-to-metallic phase transition via external biasing is integrated with the metasurface. Mie resonance-based multipolar interaction in metasurfaces hybridized with VO₂ thin film provides the difference in absorption under the opposite directions of illumination. The experimental validation of nonreciprocal light transmission through such a structure is implemented at low-level input intensities. Considering the importance of achieving nonreciprocity in compact nanophotonic structures compatible with photonic integrated circuits, I believe this paper should be of great interest to the broad readership of Nature Communications, and support its publication after the following comments are addressed:

1. The diameters of the silicon nanopillars are not given in the manuscript. Besides, as far as I inspect from the SEM image, the uniformity of the pillars is not perfect. This is not surprising. However, there should be some comment about the consequences of undesired fabrication-related problems.
2. The abstract contains weak statements. "The resonant response of the metasurface enhances nonreciprocity." The type of the resonance behavior of the metasurface should be mentioned. "The phase transition is induced all-optically by an incident light, and it is designed to occur under different conditions for the opposite directions of illumination." That statement should mention the necessity to bias the device at a proper temperature level. Besides, the targeted operating wavelength region is not specified.
3. "Nanoresonators assembled into two-dimensional lattices– metasurfaces – enabled the miniaturization of functional optical components up to the nanoscale." It's not the only wave of realizing 2D metasurfaces. Please modify the sentence to cover other approaches to implement metasurfaces.
4. The role of the VO₂ thin film thickness is not clear from the manuscript.
5. "We note that at all the temperatures the metasurface behaves similarly and requires similar levels of incident power." That statement does not completely convey the measured data. The hysteresis behavior changes (the sharpness of the transition as well as the transition points) as the bias temperature varies. This observation should be clarified instead of ignoring that response.
6. The selection of 1470 nm rather than 1550 nm?
7. Figure 3: The marker description is missing in the caption.
8. Authors make a statement: "MQ mode is more tightly localized spatially leading to higher field concentration inside the VO₂ film, and thus to higher absorption." E/H field plots numerically calculated can be used to support this explanation.
9. The selection of VO₂ compared to other types of phase change materials should be specified.
10. "Figure 4b shows a peculiar example of four distinct levels of transmission at the same wavelength and the same incident intensity..." That description corresponds to Fig. 4c.
11. "Our hybrid metasurface demonstrates over 100 nm bandwidth in the vicinity of 1.5 μm wavelength." What is the reason to have wide bandwidth in spite of the Mie-resonance response being a narrow band?
12. Lastly, authors should provide brief comments about the tolerance of their design under small oblique incidence cases.

SUMMARY OF CHANGES AND POINT-BY-POINT RESPONSES TO THE REVIEWERS' COMMENTS

Below, we provide a summary of changes to the major revision of our manuscript in the form of the point-by-point response to the referee's comments. The central major addition to our manuscript is a careful analysis of time dynamics of our metasurface. We report switching times down to picosecond range, which is ground-breaking for a nanoscale nonreciprocal component. This is enabled by ultra-fast switching time of the enabling material – VO₂ – known for its picosecond-scale insulator-to-metal transition [PNAS **114**, 9558 (2017)]. This key new addition to the manuscript serves as a demonstration of the potential of our metasurface also for ultra-fast photonics applications.

The experimental demonstration of such ultrathin nonreciprocal nanoscale structure with exceptional nonreciprocal transmission performance is unique and would be very interesting to the broad audience of Nature Communications.

REPLY TO REVIEWERS' COMMENTS

Reply to Reviewer #1

The authors demonstrate nonreciprocal transmission through an ultrathin (half-a-micron-thick) silicon metasurface hybridized with an even thinner (35nm thick) vanadium dioxide (VO₂) layer. Reciprocity is broken by VO₂ undergoing a phase transition from its insulating to conductive phase. Hence, the nonreciprocal effect is based on the thermal phase transition of VO₂ boosted by the Mie resonant response of the dielectric metasurface. The phase transition is induced all-optically by the incident light, and it is designed to occur under different conditions for the opposite directions of illumination.

The authors have adequately responded to the majority of the previous reviewers' comments. While it is true that some parts of theory and relevant nonreciprocal concepts have been presented in other works, I strongly agree with the reviewer's 2 opinion that the paper should be published after some revisions.

Our reply:

We thank the referee for their recommendation to publish our manuscript

1) Such **broadband, low power, and nanoscale ultrathin nonreciprocal design** is commendable and unique feature of this work. In addition, the effect happens in a self-biased way, i.e., by the signal itself travelling through its structure. **Hence, this modulation technique is more simple to implement compared to magnets or time-modulation and requires reduced power.** *I suggest all these major advantages of the presented system to be further stressed in the revised paper, even in the abstract section, which currently seems to be too technical for a broad audience paper.*

Our reply:

We have modified the abstract in line with the referee's suggestions

2) Since the presented thermal nonreciprocity occurs in a relatively slow time (on the order of 1 millisecond), I suggest the authors to consider adding a simulation plot similar to figure 5b in ref [34] (10.48550/arxiv.2210.05586) that demonstrates how the temperature is increased in the sample as a function of time. I was also wondering if there is a way to either experimentally or theoretically demonstrate how the

nonreciprocal transmission varies as a function of time in the case of front and back side illumination during the phase transition of VO₂.

Our reply:

Our updated manuscript now has new theoretical data and associated discussions on time dynamics of the non-reciprocal transmission. We note that our prior comment on a millisecond timescale was due to our insufficient research into the topic. Since then, we became aware of experimental observations of pico-second scale insulator-to-metal transition of the VO₂ material [Proc Natl Acad Sci U S A, 114, 36, 9558–9563, (2017)]. Our new calculations agree well with prior experimental reports demonstrating transmission fall time (due to heating) on the order of 100 ps and transmission rise time (due to cooling) on the sub-microsecond scale.

3) Relevant to my previous comment, a theoretical or experimental plot that demonstrates the nonreciprocal transmission variation in the case of front and back side illumination but now, as a function of temperature will be very interesting. It will demonstrate what happens during the phase transition of VO₂, similar to the theoretical plot (fig. 3a) in this paper: Optics Express Vol. 30, Issue 22, pp. 39716-39724 (2022)

Our reply:

We now add a transmission vs temperature plot to the supplementary (for heating) – Figure S5.

We note that temperature alone does not break nonreciprocity, and therefore at low intensities transmission for the forward and backward directions remains the same.

We present supplementary figure S4 containing data on forward/backward transmission behavior versus both temperature and incident power.

4) The paper presents a passive nonlinear-based self-induced nonreciprocal system. However, efforts are also underway to further improve the performance of these systems by incorporating active (gain) media combined with nonlinear effects. As an example, active asymmetric systems based on the coexistence of nonlinearity and parity–time symmetry can further improve the nonreciprocal transmission contrast (check for example the following papers: i) Nature Phys. 10, 394–398 (2014), ii) Nature Photonics volume 8, pages 524–529 (2014), iii) Adv. Opt. Mater. 7, 1901083 (2019).

I suggest adding some comments in the introduction relevant to active nonreciprocal nonlinear systems.

Our reply:

We have included references to PT-symmetric systems exhibiting nonreciprocity with a corresponding description.

5) Please doublecheck the entire paper and supplementary for typos. As an example, the units (W/cm²) are missing in the x-axis of Figure S4 in the supplementary document.

Our reply:

Figure S4 was fixed along with other typos that we were able to identify.

Reply to Reviewer #2

The manuscript entitled “Nanoscale optical nonreciprocity with nonlinear metasurfaces” by Aditya Tripathi et. al. presented an a-Si on VO₂ thin film nonreciprocal metasurface leveraging the out-of-plane anisotropic resonance scattering to induce different modal overlap with the absorptive VO₂ thin film between the forward and backward propagating beams. The absorption further induced the VO₂ phase-transition to metallic state (nonlinearity), achieving nonreciprocal light transmission.

My biggest concern is lack of novelty. The concept is well known and there are already several reports on such a-Si/VO₂ hybrid structure. Some examples are:

Photon. Res. 10, 373-380 (2022),

Conference on Lasers and Electro-Optics, Technical Digest Series (Optica Publishing Group, 2022), paper FTh4D.1.

An advancement to the previous work done by the authors' group (ACS Photonics 2021, 8, 4, 1206–1213) is merely changing the phase-transition stimulus from direct heating to laser absorption. The structure, and even the results are quite similar.

Our reply:

We believe this statement contains a factual error. None of the aforementioned works report nonreciprocity and as such they do not undermine our contribution.

Next, the authors failed to present a convincing implementation scenario of such structure. While they stated in the manuscript LiDARs, optical switches are possible. LiDARs are a bi-directionally transmitting case where such nonlinearity induced nonreciprocity fails. They also require phase manipulation, an important functionality this simple meta-structure does not have.

Our reply:

We strongly disagree with this opinion. Nonlinearity-based nonreciprocity (albeit on a larger scale) has already been demonstrated to be well-suited for LiDAR applications (see e.g. Nature Photonics 14, 369–374 (2020) – ref. [17] in the manuscript). At the same time, Huygens metasurfaces (underlying principle of the operation) have already been demonstrated for phase manipulation, e.g.:

- “High-Efficiency Dielectric Huygens’ Surfaces”, Adv. Optical Mater. 3, 813–820 (2015)
- “Grayscale transparent metasurface holograms”, Optica 3, 1504-1505 (2016)
- “Dielectric Broadband Metasurfaces for Fiber Mode-Multiplexed Communications”, Adv. Optical Mater. 2019, 1801679

Optical switching of ms speed is also way too slow for high-speed routing.

Our reply:

we thank the reviewer for this criticism which inspired further research resulting in new data presented at the manuscript. While the former statement on ms speed was a reference to prior research on thicker, unstructured VO₂ films, we now show that our nanostructure, with an ultra-thin VO₂ layer, has switching times down to the picosecond range.

In addition, the performance of this device is very limited. For both on-chip and free space nonreciprocal transmission, more than 10 dB isolation ratio are reported. However, in this work, the largest ratio is around 5. Though theoretically predicted a ratio of 80, its huge difference with experimental demonstration are not discussed.

Our reply:

We have added discussions on these points. The fabrication of VO₂ integrated nano-structures, especially those that utilize reactive ion etching (as this does) are in their infancy. There is much work to be done to preserve VO₂ contrast when reactive gases and additional temperature cycling, during fabrication, is present. These are, however, technical challenges, that can be solved with further engineering development.

Overall, due to insufficient novelty and limited advancement compared to previous works, I cannot recommend its publication in Nature Communications.

Reply to Reviewer #3

A. Tripathi et.al., have explored a metasurface made of Silicon nanopillars supporting Mie resonances. Thin film VO₂ exhibiting reversible insulator-to-metallic phase transition via external biasing is integrated with the metasurface. Mie resonance-based multipolar interaction in metasurfaces hybridized with VO₂ thin film provides the difference in absorption under the opposite directions of illumination. The experimental validation of nonreciprocal light transmission through such a structure is implemented at low-level input intensities. Considering the importance of achieving nonreciprocity in compact nanophotonic structures compatible with photonic integrated circuits, I believe this paper should be of great interest to the broad readership of Nature Communications and support its publication after the following comments are addressed:

1. The diameters of the silicon nanopillars are not given in the manuscript. Besides, as far as I inspect from the SEM image, the uniformity of the pillars is not perfect. This is not surprising. However, there should be some comment about the consequences of undesired fabrication-related problems.

Our reply:

We now list the geometrical parameters of the metasurface and have added a comment on fabrication imperfections:

“The metasurface consists of silicon disks 540 nm in height and diameter residing on 35-nm-thin VO₂ film arranged into a square lattice with 820 nm period”

“We attribute discrepancies between theory and experiment to imperfections associated with nanofabrication of the VO₂ and silicon-based nanostructures. The use of reactive ion etching likely causes some damage in the VO₂, reducing contrast in switching. This could be avoided by employing an architecture where VO₂ is deposited after patterning of the silicon resonator layer.”

2. The abstract contains weak statements. “The resonant response of the metasurface enhances nonreciprocity.” The type of the resonance behavior of the metasurface should be mentioned.

Our reply:

we now specify in the abstract:

“The effect is governed by the magneto-electric coupling between Mie modes supported by the resonator.”

“The phase transition is induced all-optically by an incident light, and it is designed to occur under different conditions for the opposite directions of illumination.” That statement should mention the necessity to bias the device at a proper temperature level.

Our reply:

temperature biasing is, in fact, unnecessary

We clarify in the main text: *“We study the performance of the metasurface for different temperature biases starting from room temperature (see Figure S4). We monitor the temperatures with a controller Thorlabs TC300. We observe nonreciprocal behaviour of the metasurface for all the temperature biases. We further choose to work at 40°C temperature which requires 30% less incident power to trigger the VO₂ transition when compared to room-temperature operation, while keeping the temperature sufficiently far from hysteresis of the VO₂ film.”*

Besides, the targeted operating wavelength region is not specified.

Our reply:

We now specify in the abstract: “Nonreciprocal transmission is broadband covering over 100 nm of spectrum in the telecommunication range in the vicinity of $\lambda=1.5 \mu\text{m}$ wavelength.”

3. “Nanoresonators assembled into two-dimensional lattices– metasurfaces – enabled the miniaturization of functional optical components up to the nanoscale.” It’s not the only wave of realizing 2D metasurfaces. Please modify the sentence to cover other approaches to implement metasurfaces.

Our reply:

We reformulate: “*Here we demonstrate free-space nonreciprocal transmission through a metasurface consisting of a two-dimensional layout of nanoresonators made of silicon hybridized with vanadium dioxide (VO₂).*”

4. The role of the VO₂ thin film thickness is not clear from the manuscript.

Our reply:

We now comment: “*Geometrical parameters of the metasurface including the VO₂ thickness were optimized to maximize (i) transmission in forward direction and (ii) isolation in backward direction.*”

5. “We note that at all the temperatures the metasurface behaves similarly and requires similar levels of incident power.” That statement does not completely convey the measured data. The hysteresis behavior changes (the sharpness of the transition as well as the transition points) as the bias temperature varies. This observation should be clarified instead of ignoring that response.

Our reply:

We reformulate: “*We observe nonreciprocal behaviour of the metasurface for all the temperature biases.*”

6. The selection of 1470 nm rather than 1550 nm?

Our reply:

Indeed, 1550 nm wavelength is an attractive choice as it falls into optical fibre C-band which possesses the lowest attenuation, while 1470 nm falls into the adjacent S-band.

In our experiments, we fabricated multiple samples with various size biases bearing in mind imperfections of the fabrication process. We observed best performance for samples working in the 1470 nm range, which is a 5% deviation in wavelength from 1550 nm.

While it is beyond any doubt that the fabrication can be adjusted to offset wavelength by 5%, we chose not to process additional samples as we believe that demonstration of C-band operation versus S-band operation would not increase the value of our research results, which emphasize fundamental aspects.

7. Figure 3: The marker description is missing in the caption.

Our reply:

This was fixed.

8. Authors make a statement: “MQ mode is more tightly localized spatially leading to higher field concentration inside the VO₂ film, and thus to higher absorption.” E/H field plots numerically calculated can be used to support this explanation.

Our reply:

We added E-field distributions as supplementary figure S6 with the reference in the main text.

9. The selection of VO₂ compared to other types of phase change materials should be specified.

Our reply:

Our manuscript mentions that:

“VO₂ is a particularly attractive phase-transition material whose dynamical change of phase corresponds to a subtle crystalline-to-crystalline transition and is therefore fully reversible. The exceptionally large complex refractive index variation produced by the insulator-to-conductor transition of this material made it an attractive choice for metasurfaces reconfigurable thermally or electrically [36-41]. VO₂ insulator-to-conductor phase transition was demonstrated to occur on a picosecond scale paving the way to ultra-fast applications [63].”

We however believe, that nonreciprocity in metasurfaces loaded with other types of phase-change / phase-transition materials should be studied as well. We now comment in the text:

“We note however that the principles of operation of our metasurface should be immediately applicable to other types of phase-change / phase-transition materials [64] notably including GST materials in which femto-seconds scale switching times have been reported [65].”

And in the conclusion:

“The basic principles of operation of our nonreciprocal metasurface should be immediately applicable to other types of phase-change / phase-transition materials.”

10. “Figure 4b shows a peculiar example of four distinct levels of transmission at the same wavelength and the same incident intensity...” That description corresponds to Fig. 4c.

Our reply:

This was fixed

11. “Our hybrid metasurface demonstrates over 100 nm bandwidth in the vicinity of 1.5 μm wavelength.” What is the reason to have wide bandwidth in spite of the Mie-resonance response being a narrow band?

Our reply:

We believe, this is due to (i) quality factors (Q-factors) of individual Mie resonances and (ii) multiple Mie resonances being present, thus resembling the condition of Generalized Huygens

metasurfaces. We now comment in the text that such bandwidth is not unusual for Generalized Huygens metasurfaces:

“... resembling the conditions of generalized Huygens’ principle [44]. Generalized Huygens metasurfaces are known for their extended spectral ranges of operation [44].”

12. Lastly, authors should provide brief comments about the tolerance of their design under small oblique incidence cases.

Our reply:

We now comment that: *“Metasurfaces governed by lower-order Mie resonances typically exhibit tolerance to oblique incidence angles of up to few degrees [66]. We estimate that our metasurface demonstrates small changes of transmission for ± 5 degree incident angle variation with substantial changes for ± 10 degree variation (Fig. S8).”*

REVIEWERS' COMMENTS

Reviewer #1 (Remarks to the Author):

The work is very interesting and most of my comments have been addressed. It should be published after some minor revisions:

1) The addition of time dynamic simulations (new figs. 2b and 2d) are very interesting. I suggest the authors to provide some additional details on how they performed these simulations in COMSOL that prove the picosecond ultrafast response of the presented structure. Did they take into account the intermediate phase of VO₂ when changing from amorphous to crystalline and the inverse? As an example, the complex dielectric constant of VO₂ in transition mode at temperatures very close to critical was computed by using the Bruggeman effective medium theory in this paper (see Eqs. 1-2):

<https://pubs.acs.org/doi/10.1021/acs.jpcc.1c08049>

I understand that such complex model incorporation maybe a subject of future work but some additional details on the COMSOL transient simulations are strongly suggested. Moreover, some comments about a potential experimental verification to measure the nonreciprocal transmission transient response will be very beneficial.

2) Some typos need to be corrected: i) The y-axis in Figure S5 is missing.

ii) On top of page 6 in main paper '16 μm ' should be replaced with '1.6 μm '.

Reviewer #3 (Remarks to the Author):

Authors have responded well considering the comments of the all reviewers. I recommend the acceptance of the paper to be published in Nature Communications. All optically realized nonreciprocal transmission property of thin metasurface integrated with ultra-thin VO₂ layer makes the reported results important to be timely published.

REESPONSE TO REVIEWERS' COMMENTS

Reviewer #1 (Remarks to the Author):

The work is very interesting and most of my comments have been addressed. It should be published after some minor revisions:

1) The addition of time dynamic simulations (new figs. 2b and 2d) are very interesting. I suggest the authors to provide some additional details on how they performed these simulations in COMSOL that prove the picosecond ultrafast response of the presented structure. Did they take into account the intermediate phase of VO₂ when changing from amorphous to crystalline and the inverse? As an example, the complex dielectric constant of VO₂ in transition mode at temperatures very close to critical was computed by using the Bruggeman effective medium theory in this paper (see Eqs. 1-2):

<https://pubs.acs.org/doi/10.1021/acs.jpcc.1c08049>

I understand that such complex model incorporation maybe a subject of future work but some additional details on the COMSOL transient simulations are strongly suggested. Moreover, some comments about a potential experimental verification to measure the nonreciprocal transmission transient response will be very beneficial.

Our reply:

We now added a more detailed description of our calculations and the associated references. We indeed used the Bruggeman effective medium theory. Our description reads:

We utilized the Finite Element Method (FEM) with Comsol Multiphysics® software to analyze the thermal and optical characteristics of the proposed VO₂ metasurface. Port boundary conditions facilitated the launch of transverse magnetic (TM) or transverse electric (TE) polarized plane waves, while periodic boundary conditions simulated the side boundaries. Although lasers typically exhibit Gaussian beam profiles, the simulation geometry's small size enabled us to treat the laser as a plane wave. By combining the Heat Transfer in Solids and Electromagnetic Waves modules, we modeled transient heating and calculated transmittance over time. Specifically, we examined the transmission under front and back side illumination at 1.44 μm wavelength, with an input power intensity of 22.3 kW/cm² and a pulse duration of 100 ps (assuming a simplistic rectangular pulse shape). The initial temperature was set to room temperature (293K) and was uniform throughout the structure. Electromagnetic pulse in COMSOL introduced a heat source term based on optical absorption of materials calculated within the Electromagnetic Waves module. Temperature-dependent thermal conductivity, heat capacity, and density values of materials PMMA, Al₂O₃, and SiO₂ in the thermal simulations were taken from experimental data⁴⁷⁻⁴⁹. Meanwhile, the VO₂ dielectric constant was determined using the Bruggeman effective medium theory⁵⁰

$$\epsilon_{VO_2} = \frac{1}{4} \left[\epsilon_i(2 - 3V) + \epsilon_c(3V - 1) + \sqrt{[\epsilon_i(2 - 3V) + \epsilon_c(3V - 1)]^2 + 8\epsilon_i\epsilon_c} \right] \quad (1)$$

the ϵ_i and ϵ_c are represented by the dielectric constants of the insulating and conducting phases of VO₂, respectively. They have been measured experimentally. The metallic volume fraction, indicated by V, can be calculated using the following formula:

$$V = 1 - \frac{1}{1 + e^{\frac{T - T_c}{\Delta T}}} \quad (2)$$

T is the ambient temperature, T_c is the critical temperature of VO₂, and ΔT denotes the transition width and equals 2K⁵¹. The optical properties of VO₂ are incorporated into an electromagnetic simulation to analyze the metasurface's time-dependent transmittance and temperature. The transmission coefficient was calculated from S-parameters and temperature was integrated over the volume of the VO₂ film.

2) Some typos need to be corrected: i) The y-axis in Figure S5 is missing.

Our reply:

Corrected

ii) On top of page 6 in main paper '16 μm ' should be replaced with '1.6 μm '.

Our reply:

Corrected

Reviewer #3 (Remarks to the Author):

Authors have responded well considering the comments of the all reviewers. I recommend the acceptance of the paper to be published in Nature Communications. All optically realized nonreciprocal transmission property of thin metasurface integrated with ultra-thin VO₂ layer makes the reported results important to be timely published.

Our reply:

We thank the referee for the positive evaluation of our work